# Can SGD Learn Recurrent Neural Networks with Provable Generalization?*

**Zeyuan Allen-Zhu**
Microsoft Research AI
zeyuan@csail.mit.edu

**Yuanzhi Li**
Carnegie Mellon University
yuanzhil@andrew.cmu.edu

## Abstract

Recurrent Neural Networks (RNNs) are among the most popular models in sequential data analysis. Yet, in the foundational PAC learning language, what concept class can it learn? Moreover, how can the same recurrent unit simultaneously learn functions from different input tokens to different output tokens, without affecting each other? Existing generalization bounds for RNN scale exponentially with the input length, significantly limiting their practical implications.

In this paper, we show using the vanilla stochastic gradient descent (SGD), RNN can actually learn some notable concept class *efficiently*, meaning that both time and sample complexity scale *polynomially* in the input length (or almost polynomially, depending on the concept). This concept class at least includes functions where each output token is generated from inputs of earlier tokens using a smooth two-layer neural network.

## 1   Introduction

Recurrent neural networks (RNNs) is one of the most popular models in sequential data analysis [25]. When processing an input sequence, RNNs repeatedly and sequentially apply the same operation to each input token. The recurrent structure of RNNs allows it to capture the dependencies among different tokens inside each sequence, which is empirically shown to be effective in many applications such as natural language processing [28], speech recognition [12] and so on.

The recurrent structure in RNNs shows great power in practice, however, it also imposes great challenge in theory. Until now, RNNs remains to be one of the least theoretical understood models in deep learning. Many fundamental open questions are still largely unsolved in RNNs, including

1. (Optimization). When can RNNs be trained efficiently?
2. (Generalization). When do the results learned by RNNs generalize to test data?

Question 1 is technically challenging due to the notorious question of vanishing/exploding gradients, and the non-convexity of the training objective induced by *non-linear* activation functions.

Question 2 requires even deeper understanding of RNNs. For example, in natural language processing, "*Juventus* beats Bacerlona" and "Bacerlona beats *Juventus*" have completely different meanings. How can the same operation in RNN encode a *different rule* for "Juventus" at token 1 vs. "Juventus" at token 3, instead of merely memorizing each training example?

There have been some recent progress towards obtaining more principled understandings of these questions. On the optimization side, Hardt, Ma, and Recht [13] show that over-parameterization can help in the training process of a linear dynamic system, which is a special case of RNNs with *linear* activation functions. Allen-Zhu, Li, and Song [2] show that over-parameterization also helps in training RNNs with ReLU activations. This latter result gives no generalization guarantee.

On the generalization side, our understanding to RNN is even more limited. The VC-dimension bounds [10] and [17] polynomially depend on the size of the network, and either only apply to linear (or threshold) networks or to networks with one dimension input. However, a bound scaling with the total number of parameters usually cannot be applied to modern neural networks, which are heavily over-parameterized. Others [9, 31] (or the earlier work [14]) establish sample complexity bounds that exponentially grow in the input length. In particular, they depend on the operator norm of the recurrent unit, that we denote by $\beta$. If $\beta > 1$, their bounds scale *exponentially* with input length. Since most applications do not regularize $\beta$ and allow $\beta > 1$ for a richer expressibility,[2] their bounds are still insufficient.

Indeed, bridging the gap between optimization (question 1) and generalization (question 2) can be quite challenging in neural networks. The case of RNN is particularly so due to the (potentially) exponential blowup in input length.

- Generalization $\nrightarrow$ Optimization. One could imagine adding a strong regularizer to ensure $\beta \leq 1$ for generalization purpose; however, it is unclear how an optimization algorithm such as stochastic gradient descent (SGD) finds a network that *both* minimizes training loss and maintains $\beta \leq 1$. One could also use a very small network so the number of parameters is limited; however, it is not clear how SGD finds a small network with small training loss.

- Optimization $\nrightarrow$ Generalization. One could try to train RNNs without any regularization; however, it is then quite possible that the number of parameters need to be large and $\beta > 1$ after the training. This is so both in practice (since "memory implies larger spectral radius" [24]) and in theory [2]. All known generalization bounds fail to apply in this regime.

In this paper, we give arguably the first theoretical analysis of RNNs that captures optimization and generalization *simultaneously*. Given any set of input sequences, as long as the outputs are (approximately) realizable by some smooth function in a certain concept class, then after training a vanilla RNN with ReLU activations, SGD provably finds a solution that has both small training and generalization error. Our result allows $\beta$ to be *larger* than 1 by a constant, but is still *efficient*: meaning that the iteration complexity of the SGD, the sample complexity, and the time complexity scale only *polynomially* (or almost polynomially) with the length of the input.

## 2 Notations

We denote by $\|\cdot\|_2$ (or sometimes $\|\cdot\|$) the Euclidean norm of vectors, and by $\|\cdot\|_2$ the spectral norm of matrices. We denote by $\|\cdot\|_\infty$ the infinite norm of vectors, $\|\cdot\|_0$ the sparsity of vectors or diagonal matrices, and $\|\cdot\|_F$ the Frobenius norm of matrices. Given matrix $W$, we denote by $W_k$ or $w_k$ the $k$-th row vector of $W$. We denote the row $\ell_p$ norm for $W \in \mathbb{R}^{m \times d}$ as $\|W\|_{2,p} := \left( \sum_{i \in [m]} \|w_i\|_2^p \right)^{1/p}$. By definition, $\|W\|_{2,2} = \|W\|_F$. We use $\mathcal{N}(\mu, \sigma)$ to denote Gaussian distribution with mean $\mu$ and variance $\sigma$; or $\mathcal{N}(\mu, \Sigma)$ to denote Gaussian vector with mean $\mu$ and covariance $\Sigma$. We use $x = y \pm z$ to denote that $x \in [y - z, y + z]$. We use $\mathbb{1}_{event}$ to denote the indicator function of whether $event$ is true. We denote by $\mathbf{e}_k$ the $k$-th standard basis vector. We use $\sigma(\cdot)$ to denote the ReLU function $\sigma(x) = \max\{x, 0\} = \mathbb{1}_{x \geq 0} \cdot x$. Given univariate function $f \colon \mathbb{R} \to \mathbb{R}$, we also use $f$ to denote the same function over vectors: $f(x) = (f(x_1), \ldots, f(x_m))$ if $x \in \mathbb{R}^m$.

Given vectors $v_1, \ldots, v_n \in \mathbb{R}^m$, we define $U = \mathsf{GS}(v_1, \ldots, v_n)$ as their Gram-Schmidt orthonormalization. Namely, $U = [\widehat{v}_1, \ldots, \widehat{v}_n] \in \mathbb{R}^{m \times n}$ where $\widehat{v}_1 = \frac{v_1}{\|v_1\|}$ and for every $i \geq 2$, $\widehat{v}_i = \frac{\prod_{j=1}^{i-1}(I - \widehat{v}_j \widehat{v}_j^\top) v_i}{\|\prod_{j=1}^{i-1}(I - \widehat{v}_j \widehat{v}_j^\top) v_i\|}$. Note that in the occasion that $\prod_{j=1}^{i-1}(I - \widehat{v}_j \widehat{v}_j^\top) v_i$ is the zero vector, we let $\widehat{v}_i$ be an arbitrary unit vector that is orthogonal to $\widehat{v}_1, \ldots, \widehat{v}_{i-1}$.

We say a function $f \colon \mathbb{R}^d \to \mathbb{R}$ is $L$-Lipscthiz continuous if $|f(x) - f(y)| \leq L\|x - y\|_2$; and say it is is $L$-smooth if its gradient is $L$-Lipscthiz continuous, that is $\|\nabla f(x) - \nabla f(y)\|_2 \leq L\|x - y\|_2$.

**Function complexity.** The following notions from [1] measure the complexity of any infinite-order smooth function $\phi \colon \mathbb{R} \to \mathbb{R}$. Suppose $\phi(z) = \sum_{i=0}^\infty c_i z^i$ is its Taylor expansion. Given

non-negative $R$,

$$\mathfrak{C}_\varepsilon(\phi, R) := \sum_{i=0}^\infty \left( (C^* R)^i + \left( \frac{\sqrt{\log(1/\varepsilon)}}{\sqrt{i}} C^* R \right)^i \right) |c_i|$$

$$\mathfrak{C}_{\mathfrak{s}}(\phi, R) := C^* \sum_{i=0}^\infty (i+1)^{1.75} R^i |c_i|$$

where $C^*$ is a sufficiently large constant (e.g., $10^4$). It holds $\mathfrak{C}_{\mathfrak{s}}(\phi, R) \leq \mathfrak{C}_\varepsilon(\phi, R) \leq \mathfrak{C}_{\mathfrak{s}}(\phi, O(R)) \times$ poly$(1/\varepsilon)$, and for $\sin z, e^z$ or low degree polynomials, they only differ by $o(1/\varepsilon)$. [1]

**Example 2.1.** If $\phi(z) = z^d$ for constant $d$ then $\mathfrak{C}_{\mathfrak{s}}(\phi, R) \leq O(R^d)$, $\mathfrak{C}_\varepsilon(\phi, R) \leq O(R^d \text{polylog}(\frac{1}{\varepsilon}))$. For functions such as $\phi(z) = e^z - 1, \sin z$, sigmoid$(z)$ or $\tanh(z)$, it suffices to consider $\varepsilon$-approximations of them so we can truncate their Taylor expansions to degree $O(\log(1/\varepsilon))$. This gives $\mathfrak{C}_{\mathfrak{s}}(\phi, R), \mathfrak{C}_\varepsilon(\phi, R) \leq (1/\varepsilon)^{O(\log R)}$.

# 3 Problem Formulation

The data are generated from an unknown distribution $\mathcal{D}$ over $(x^\star, y^\star) \in (\mathbb{R}^{d_x})^{(L-2)} \times \mathcal{Y}^{(L-2)}$. Each input sequence $x^\star$ consists of $x_2^\star, \ldots, x_{L-1}^\star \in \mathbb{R}^{d_x}$ with $\|x_\ell^\star\| = 1$ and $[x_\ell^\star]_{d_x} = \frac{1}{2}$ without loss of generality.[3] Each label sequence $y^\star$ consists of $y_3^\star, \ldots, y_L^\star \in \mathcal{Y}$. The training dataset $\mathcal{Z} = \{((x^\star)^{(i)}, (y^\star)^{(i)})\}_{i \in [N]}$ is given as $N$ i.i.d. samples from $\mathcal{D}$. When $(x^\star, y^\star)$ is generated from $\mathcal{D}$, we call $x^\star$ the *true* input sequence and $y^\star$ the true label.

**Definition 3.1.** *Without loss of generality (see Remark 3.4), for each true input $x^\star$, we transform it into an* actual input *sequence* $x_1, x_2, \ldots, x_L \in \mathbb{R}^{d_x+1}$ *as follows.*

$$x_1 = (0^{d_x}, 1) \quad and \quad x_\ell = (\varepsilon_x x_\ell^\star, 0) \quad for \quad \ell = 2, 3, \ldots, L-1 \quad and \quad x_L = (0^{d_x}, \varepsilon_x)$$

*where $\varepsilon_x \in (0, 1)$ is a parameter to be chosen later. We then feed this actual sequence $x$ into RNN.*

**Definition 3.2.** *We say the sequence $x_1, \ldots, x_L \in \mathbb{R}^{d_x+1}$ is* normalized *if*

$$\|x_1\| = 1 \quad and \quad \|x_\ell\| = \varepsilon_x \quad for \ all \ \ell = 2, 3, \ldots, L.$$

## 3.1 Our Learner Network: Elman RNN

To present the simplest result, we focus on the classical Elman RNN with ReLU activation. Let $W \in \mathbb{R}^{m \times m}$, $A \in \mathbb{R}^{m \times (d_x+1)}$, and $B \in \mathbb{R}^{d \times m}$ be the weight matrices.

**Definition 3.3.** *Our Elman RNN can be described as follows. On input $x_1, \ldots, x_L \in \mathbb{R}^{d_x+1}$,*

$$h_0 = 0 \in \mathbb{R}^m \qquad\qquad g_\ell = W \cdot h_{\ell-1} + A x_\ell \in \mathbb{R}^m$$

$$y_\ell = B \cdot h_\ell \in \mathbb{R}^d \qquad\qquad h_\ell = \sigma(W \cdot h_{\ell-1} + A x_\ell) \in \mathbb{R}^m$$

*We say that $W, A, B$ are at* random initialization, *if the entries of $W$ and $A$ are i.i.d. generated from $\mathcal{N}(0, \frac{2}{m})$, and the entries of $B$ are i.i.d. generated from $\mathcal{N}(0, \frac{1}{d})$.*

For simplicity, in this paper we only update $W$ and let $A$ and $B$ be at their random initialization. Thus, we write $F_\ell(x^\star; W) = y_\ell = B h_\ell$ as the output of the $\ell$-th layer.

Our goal is to use $y_3, \ldots, y_L \in \mathbb{R}^d$ to fit the true label $y_3^\star, \ldots, y_L^\star \in \mathcal{Y}$ using some *loss* function $G \colon \mathbb{R}^d \times \mathcal{Y} \to \mathbb{R}$. In this paper we assume, for every $y^\star \in \mathcal{Y}$, $G(0^d, y^\star) \in [-1, 1]$ is bounded, and $G(\cdot, y^\star)$ is convex and 1-Lipschitz continuous in its first variable. This includes for instance the cross-entropy loss and $\ell_2$-regression loss (for $y^\star$ being bounded).[4]

*Remark* 3.4. Since we only update $W$, the label sequence $y_3^\star, \ldots, y_L^\star$ is off from the input sequence $x_2^\star, \ldots, x_{L-1}^\star$ by one. The last $x_L$ can be made zero, but we keep it normalized for notational simplicity. The first $x_1$ gives a random seed fed into the RNN (one can equivalently put it into $h_0$). We have scaled down the input signals by $\varepsilon_x$, which can be equivalently thought as scaling down $A$.

## 3.2 Concept Class

Let $\{\Phi_{i \to j, r, s} \colon \mathbb{R} \to \mathbb{R}\}_{i,j \in [L], r \in [p], s \in [d]}$ be infinite-order differentiable functions, and $\{w_{i \to j, r, s}^* \in \mathbb{R}^{d_x}\}_{i,j \in [L], r \in [p], s \in [d]}$ be unit vectors. Then, for every $j = 3, 4, \ldots, L$, we consider *target functions* $F_j^* \colon \mathbb{R}^{d_x} \to \mathbb{R}^d$ where $F_j^* = (F_{j,1}^*, \ldots, F_{j,d}^*)$ can be written as

$$F_{j,s}^*(x^\star) := \textstyle\sum_{i=2}^{j-1} \sum_{r \in [p]} \Phi_{i \to j, r, s}(\langle w_{i \to j, r, s}^*, x_i^\star \rangle) \in \mathbb{R} \ . \tag{3.1}$$

For proof simplicity, we assume $\Phi_{i \to j, r, s}(0) = 0$. We also use

$$\mathfrak{C}_\varepsilon(\Phi, R) = \max_{i,j,r,s} \{\mathfrak{C}_\varepsilon(\Phi_{i \to j, r, s}, R)\} \quad \text{and} \quad \mathfrak{C}_\mathfrak{s}(\Phi, R) = \max_{i,j,r,s} \{\mathfrak{C}_\mathfrak{s}(\Phi_{i \to j, r, s}, R)\}$$

to denote the complexity of $F^*$.

**Agnostic PAC-learning language.** Our concept class consists of all functions $F^*$ in the form of (3.1) with complexity bounded by threshold $C$ and parameter $p$ bounded by threshold $p_0$. Let $\mathsf{OPT}$ be the population risk achieved by the *best* target function in this concept class. Then, our goal is to learn this concept class with population risk $\mathsf{OPT} + \varepsilon$ using sample and time complexity *polynomial* in $C$, $p_0$ and $1/\varepsilon$. In the remainder of this paper, to simplify notations, we do not explicitly define this concept class parameterized by $C$ and $p$. Instead, we equivalently state our theorem with respect to any (unknown) target function $F^*$ with specific parameters $C$ and $p$.

**Example 3.5.** Our concept class is general enough and contains functions where the output at each token is generated from inputs of previous tokens using any two-layer neural network. Indeed, one can verify that our general form (3.1) includes functions of the following:

$$F_j^*(x^\star) = \textstyle\sum_{i=2}^{j-1} A_{j-i}^* \phi(W_{j-i}^* x_i^\star) \ .$$

**Example 3.6.** Counting is an example task that falls into our concept class. Specifically, one can define $\phi$ such that $\phi(a) = 1$ and $\phi(b) = -1$ (this can be achieved by a quadratic function with constant complexity). The target function can be $\sum_i \phi(x_i)$. If the sequence is $x = a^n b^m$ such that $n = m$, then $\sum_i \phi(x_i) = 0$, otherwise it is non-zero.

# 4  Our Result: RNN Provably Learns the Concept Class

Suppose the distribution $\mathcal{D}$ is generated by some (unknown) target function $F^*$ of the form (3.1) in the concept class with population risk $\mathsf{OPT}$, namely,

$$\mathbb{E}_{(x^\star, y^\star) \sim \mathcal{D}} \left[ \textstyle\sum_{j=3}^L G\left(F_j^*(x^\star), y_j^\star\right) \right] \le \mathsf{OPT} \ ,$$

and suppose we are given training dataset $\mathcal{Z}$ consisting of $N$ i.i.d. samples from $\mathcal{D}$. We consider the following stochastic training objective

$$\mathsf{Obj}(W') := \mathbb{E}_{(x^\star, y^\star) \sim \mathcal{Z}} \left[ \mathsf{Obj}(x^\star, y^\star; W') \right]$$

$$\text{where} \quad \mathsf{Obj}(x^\star, y^\star; W') := \textstyle\sum_{j=3}^L G\left(\lambda F_j(x^\star; W + W'), y_j^\star\right)$$

Above, $W \in \mathbb{R}^{m \times m}$ is random initialization, $W' \in \mathbb{R}^{m \times m}$ is the additional shift, and $\lambda \in (0, 1)$ is a constant scaling factor on the network output.[5] We consider the vanilla stochastic gradient descent (SGD) algorithm with step size $\eta$. In each iteration $t = 1, 2, \ldots, T$, it updates

$$W_t = W_{t-1} - \eta \nabla_{W'} \mathsf{Obj}(x^\star, y^\star; W_{t-1}) \tag{SGD}$$

for a random sample $(x^\star, y^\star)$ from the training set $\mathcal{Z}$.

**Theorem 1.** *For every* $0 < \varepsilon < \widetilde{O}\big(\frac{1}{\mathsf{poly}(L,d) \cdot p \cdot \mathfrak{C}_\mathfrak{s}(\Phi, O(\sqrt{L}))}\big)$, *define complexity* $C = \mathfrak{C}_\varepsilon(\Phi, \sqrt{L})$ *and* $\lambda = \widetilde{\Theta}\big(\frac{\varepsilon}{L^2 d}\big)$, *if the number of neurons* $m \ge \mathsf{poly}(C, \varepsilon^{-1})$ *and the number of samples is* $N = |\mathcal{Z}| \ge \mathsf{poly}(C, \varepsilon^{-1}, \log m)$, *then SGD with* $\eta = \widetilde{\Theta}\big(\frac{1}{\varepsilon L^2 d^2 m}\big)$ *and*

$$T = \widetilde{\Theta}\Big(\frac{p^2 C^2 \mathsf{poly}(L, d)}{\varepsilon^2}\Big)$$

*satisfies that, with probability at least $1 - e^{-\Omega(\rho^2)}$ over the random initialization*

$$\mathbb{E}_{sgd}\left[\frac{1}{T}\sum_{t=0}^{T-1}\mathbb{E}_{(x^\star,y^\star)\sim\mathcal{D}}\left[\mathsf{Obj}(x^\star,y^\star;W+W_t)\right]\right]\leq \mathsf{OPT}+\varepsilon\ .$$

Above, $\mathbb{E}_{sgd}$ takes expectation with respect to the randomness of SGD. Since SGD takes only one example per iteration, the sample complexity $N$ is also bounded by $T$.

## 4.1 Our Contribution, Interpretation, and Discussion

**Sample complexity.** Our sample complexity only scales with $\log(m)$, making the result applicable to *over-parameterized* RNNs that have $m \gg N$. Following Example 2.1, if $\phi(z)$ is constant degree polynomial we have $C = \mathsf{poly}(L, \log \varepsilon^{-1})$ so Theorem 1 says that RNN learns such concept class

$$\text{with size } m = \frac{\mathsf{poly}(L,d,p)}{\mathsf{poly}(\varepsilon)} \text{ and sample complexity } \min\{N,T\} = \frac{p^2\mathsf{poly}(L,d,\log m)}{\varepsilon^2}$$

If $\phi(z)$ is a function with good Taylor truncation, such as $e^z - 1, \sin z, \mathrm{sigmoid}(z)$ or $\tanh(z)$, then $C = L^{O(\log(1/\varepsilon))}$ is almost polynomial.

**Non-linear measurements.** Our result shows that vanilla RNNs can efficiently learn a weighted average of non-linear measurements of the input. As we argued in Example 3.5, this at least includes functions where the output at each token is generated from inputs of previous tokens using any two-layer neural networks. Average of non-linear measurements can be quite powerful, achieving the state-of-the-art performance in some sequential applications such as sentence embedding [4] and many others [23], and acts as the base of attention mechanism in RNNs [5].

**Adapt to tokens.** In the target function, $\Phi_{i\to j,r,s}$ can be different at each token, meaning that they can *adapt* to the position of the input tokens. We emphasize that the positions of the tokens (namely, the values $i,j$) are not *directly fed* into the network, rather it is discovered through sequentially reading the input. As one can see from our proofs, the ability of adapting to the tokens comes from the inhomogeneity in hidden layers $h_\ell$: even when $x_\ell = x_{\ell'}$ for different tokens $\ell' \neq \ell$, there is still big difference between $h_\ell$ and $h_{\ell'}$. Albeit the *same* operator is applied to $x_\ell$ and $x_{\ell'}$, RNNs can still use this crucial inhomogeneity to learn different functions at different tokens.

In our result, the function $\Phi_{i\to j,r,s}$ only adapts with the positions of the input tokens, but in many applications, we would like the function to adapt with the values of the past tokens $x_1^\star, \dots, x_{i-1}^\star$ as well. We believe a study on other models (such as LSTM [15]) can potentially settle these questions.

**Long term memory.** It is commonly believed that vanilla RNNs cannot capture *long term* dependencies in the input. This does not contradict our result. Our complexity parameter $\mathfrak{C}_\varepsilon(\Phi, \sqrt{L})$ of the learning process in Theorem 1 indeed *suffers* from $L$, the length of the input sequence. This is due to the fact that vanilla RNN, the hidden neurons $h_\ell$ will incorporate more and more noise as the time horizon $\ell$ increases, making the new signal $Ax_\ell$ less and less significant.

**Comparison to feed-forward networks.** Recently there are many interesting results on analyzing the learning process of feed-forward neural networks [7, 8, 11, 16, 18–20, 26, 27, 29, 30, 32]. Most of them either assume that the input is structured (e.g. Gaussian or separable) or only consider linear networks. Allen-Zhu, Li, and Liang [1] show a result in the same flavor as this paper but for two and three-layer feedforward networks. Since RNNs apply the same unit repeatedly to each input token in a sequence, our analysis is significantly different from [1] and creates lots of difficulties in the analysis.[6]

### 4.2 Conclusion

We show RNN can actually learn some notable concept class *efficiently*, using simple SGD method with sample complexity polynomial or almost-polynomial in input length. This concept class at least includes functions where each output token is generated from inputs of earlier tokens using a smooth neural network. To the best of our knowledge, this is the first proof that some non-trivial concept class is *efficiently* learnable by RNN. Our sample complexity is almost independent of $m$, making the result applicable to over-parameterized settings. On a separate note, our proof explains why the same recurrent unit is capable of learning various functions from different input tokens to different output tokens.

Section 6 through 9 continue to give sketch proofs. Our final proofs reply on many other technical properties of RNN that may be of independent interests: such as properties of RNN at random initialization (which we include in Section B and C), and properties of RNN stability (which we include in Section D, E, F). Some of these properties are simple modifications from prior work, but some are *completely new* and require new proof techniques (namely, Section C, D and E).

# PROOF SKETCH

Our proof of Theorem 1 divides into four conceptual steps.

1.  We obtain first-order approximation of how much the outputs of the RNN change if we move from $W$ to $W + W'$. This change (up to small error) is a linear function in $W'$. (See Section 6).
    *(This step can be derived from prior work [2] without much difficulty.)*
2.  We construct some (unknown) matrix $W^* \in \mathbb{R}^{m \times m}$ so that this "linear function", when evaluated on $W^*$, approximately gives the target $F^*$ in the concept class (see Section 5).
    *(This step is the most interesting part of this paper.)*
3.  We argue that the SGD method moves in a direction nearly as good as $W^*$ and thus efficiently decreases the training objective (see Section 7).
    *(This is a routine analysis of SGD in the non-convex setting given Steps 1&2.)*
4.  We use the first-order linear approximation to derive a Rademacher complexity bound that does not grow exponentially in $L$ (see Section 8). By feeding the output of SGD into this Rademacher complexity, we finish the proof of Theorem 1 (see Section 9).
    *(This is a one-paged proof given the Steps 1&2&3.)*

Although our proofs are technical, to help the readers, we write 7 pages of sketch proofs for Steps 1 through 4. This can be found in Section 5 through 9. Due to space limitation, we only include Section 5 in the main body. We introduce some notations for analysis purpose.

**Definition 4.1.** *For each $\ell \in [L]$, let $D_\ell \in \mathbb{R}^{m \times m}$ be the diagonal matrix where*

$$(D_\ell)_{k,k} = \mathbb{1}_{(W \cdot h_{\ell-1} + A x_\ell)_k \geq 0} = \mathbb{1}_{(g_\ell)_k \geq 0} \ .$$

*As a result, we can write $h_\ell = D_\ell W h_{\ell-1}$. For each $1 \leq \ell \leq a \leq L$, we define*

$$\mathsf{Back}_{\ell \to a} = B D_a W \cdots D_{\ell+1} W \in \mathbb{R}^{d \times m}.$$

*with the understanding that $\mathsf{Back}_{\ell \to \ell} = B \in \mathbb{R}^{d \times m}$.*

Throughout the proofs, to simplify notations when specifying polynomial factors, we introduce

$$\rho := 100 L d \log m \quad \text{and} \quad \varrho := \frac{100 L d p \cdot \mathfrak{C}_\varepsilon(\Phi, \sqrt{L}) \cdot \log m}{\varepsilon}$$

We assume $m \geq \mathsf{poly}(\varrho)$ for some sufficiently large polynomial factor.

## 5 Existence of Good Network Through Backward

One of our main contributions is to show the existence of some "good linear network" to approximate any target function. Let us explain what this means. Suppose $W, A, B$ are at random initialization. We consider a linear function over $W^* \in \mathbb{R}^{m \times m}$:

$$f_{j'} := \sum_{i'=2}^{j'} \mathsf{Back}_{i' \to j'} D_{i'} W^* h_{i'-1} \ . \tag{5.1}$$

As we shall see later, in first-order approximation, this linear function captures how much the output of the RNN changes at token $j'$, if one we move $W$ to $W + W'$. The goal in this section is to construct some $W^{\circledast} \in \mathbb{R}^{m \times m}$ satisfying that, for *any* true input $x^{\star}$ in the support of $\mathcal{D}$, if we define the actual input $x$ according to $x^{\star}$ (see Definition 3.1), then with high probability,

$$\forall s' \in [d] \qquad f_{j',s'} \approx F^*_{j',s'}(x^{\star}) = \sum_{i=2}^{j'-1} \sum_{r \in [p]} \Phi_{i \to j',r,s'}(\langle w^*_{i \to j',r,s'}, x^{\star}_i \rangle) \qquad (5.2)$$

In our sketched proof below, it shall become clear how this *same* matrix $W^{\circledast}$ can simultaneously represent functions $\Phi_{i \to j'}$ that come from different input tokens $i$. Since SGD can be shown to descend in a direction "comparable" to $W^{\circledast}$, it converges to a matrix $W$ with similar guarantees.

## 5.1 Indicator to Function

In order to show (5.2), we first show a variant of the "indicator to function" lemma from [1].

**Lemma 5.1** (indicator to function). *For every smooth function $\Phi \colon \mathbb{R} \to \mathbb{R}$, every unit vector $w^* \in \mathbb{R}^{d_x}$ with $w^*_{d_x} = 0$, every constant $\sigma \geq 0.1$, every constant $\gamma > 1$, every constant $\varepsilon_e \in \left(0, \frac{1}{\mathfrak{C}_{\mathfrak{s}}(\Phi, O(\sigma))}\right)$, there exists*

$$C' = \mathfrak{C}_{\varepsilon_e}(\Phi, \sigma) \text{ and a function } H \colon \mathbb{R} \to [-C', C'],$$

*such that for every fixed unit vectors $x^{\star} \in \mathbb{R}^{d_x}$ with $x^{\star}_{d_x} = \frac{1}{2}$,*

*(a)* $\left| \mathbb{E}_{a \sim \mathcal{N}(0,\mathbf{I}), n \sim \mathcal{N}(0,\sigma^2)} \left[ \mathbb{1}_{\langle a, x^{\star} \rangle + n \geq 0} H(a) \right] - \Phi(\langle w^*, x^{\star} \rangle) \right| \leq \varepsilon_e$              *(on target)*

*(b)* $\left| \mathbb{E}_{a \sim \mathcal{N}(0,\mathbf{I}), n \sim \mathcal{N}(0,\sigma^2)} \left[ \mathbb{1}_{\langle a, x^{\star} \rangle + \gamma n \geq 0} H(a) \right] - \Phi(0) \right| \leq \varepsilon_e + O\left( \frac{C' \log(\gamma \sigma)}{\gamma \sigma} \right)$     *(off target)*

Above, Lemma 5.1a says that we can use a bounded function $\mathbb{1}_{\langle a, x^{\star} \rangle + n \geq 0} H(a)$ to fit a target function $\Phi(\langle w^*, x^{\star} \rangle)$, and Lemma 5.1b says that if the magnitude of $n$ is large then this function is close to being constant. For such reason, we can view $n$ as "*noise*." While the proof of 5.1a is from prior work [1], our new property 5.1b is completely new and it requires some technical challenge to simultaneously guarantee 5.1a and 5.1b. The proof is in Appendix G.1

## 5.2 Fitting a Single Function

We now try to apply Lemma 5.1 to approximate a single function $\Phi_{i \to j,r,s}(\langle w^*_{i \to j,r,s}, x^{\star}_i \rangle)$. For this purpose, let us consider two (normalized) input sequences. The first (null) sequence $x^{(0)}$ is given as

$$x^{(0)}_1 = (0^{d_x}, 1) \quad \text{and} \quad x^{(0)}_\ell = (0^{d_x}, \varepsilon_x) \text{ for } \ell = 2, 3, \dots, L$$

The second sequence $x$ is generated from an input $x^{\star}$ in the support of $\mathcal{D}$ (recall Definition 3.1). Let

- $h_\ell, D_\ell, \mathsf{Back}_{i \to j}$ be defined with respect to $W, A, B$ and input sequence $x$, and
- $h^{(0)}_\ell, D^{(0)}_\ell, \mathsf{Back}^{(0)}_{i \to j}$ be defined with respect to $W, A, B$ and input sequence $x^{(0)}$

We remark that $h^{(0)}_\ell$ has the good property that it does not depend $x^{\star}$ but somehow stays "close enough" to the true $h_\ell$ (see Appendix D for a full description).

**Lemma 5.2** (fit single function). *For every $2 \leq i < j \leq L$, $r \in [p], s \in [d]$ and every constant $\varepsilon_e \in \left(0, \frac{1}{\mathfrak{C}_{\mathfrak{s}}(\Phi_{i \to j,r,s}, O(\sqrt{L}))}\right)$, there exists $C' = \mathfrak{C}_{\varepsilon_e}(\Phi_{i \to j,r,s}, \sqrt{L})$ so that, for every*

$$\varepsilon_x \in \left(0, \frac{1}{\rho^4 C'}\right) \quad \text{and} \quad \varepsilon_c = \frac{\varepsilon_e \varepsilon_x}{4C'} \ ,$$

*there exists a function $H_{i \to j,r,s} \colon \mathbb{R} \to \left[ -\frac{4(C')^2}{\varepsilon_e \varepsilon_x}, \frac{4(C')^2}{\varepsilon_e \varepsilon_x} \right]$, such that, let*

- *$x$ be a fixed input sequence defined by some $x^{\star}$ in the support of $\mathcal{D}$ (see Definition 3.1),*
- *$W, A$ be at random initialization,*
- *$h_\ell$ be generated by $W, A, x$ and $h^{(0)}_\ell$ be generated by $W, A, x^{(0)}$, and*
- *$\widetilde{w}_k, \widetilde{a}_k \sim \mathcal{N}\left(0, \frac{2\mathbf{I}}{m}\right)$ be freshly new random vectors,*

*with probability at least $1 - e^{-\Omega(\rho^2)}$ over $W$ and $A$,*

*(a) (on target)*

$$\left| \mathbb{E}_{\widetilde{w}_k, \widetilde{a}_k} \left[ \mathbb{1}_{|\langle \widetilde{w}_k, h^{(0)}_{i-1} \rangle| \leq \frac{\varepsilon_c}{\sqrt{m}}} \mathbb{1}_{\langle \widetilde{w}_k, h_{i-1} \rangle + \langle \widetilde{a}_k, x_i \rangle \geq 0} H_{i \to j,r,s}(\widetilde{a}_k) \right] - \Phi_{i \to j,r,s}(\langle w^*_{i \to j,r,s}, x^{\star}_i \rangle) \right| \leq \varepsilon_e$$

*(b) (off target), for every $i' \neq i$*

$$\left| \underset{\widetilde{w}_k, \widetilde{a}_k}{\mathbb{E}} \left[ \mathbb{1}_{|\langle \widetilde{w}_k, h_{i-1}^{(0)} \rangle| \leq \frac{\varepsilon_c}{\sqrt{m}}} \mathbb{1}_{\langle \widetilde{w}_k, h_{i'-1} \rangle + \langle \widetilde{a}_k, x_{i'} \rangle \geq 0} H_{i \to j, r, s}(\widetilde{a}_k) \right] \right| \leq \varepsilon_e$$

Lemma 5.2 implies there is a quantity $\mathbb{1}_{|\langle \widetilde{w}_k, h_{i-1}^{(0)} \rangle| \leq \frac{\varepsilon_c}{\sqrt{m}}} H_{i \to j, r, s}(\widetilde{a}_k)$ that *only depends* on the target function and the random initialization (namely, $\widetilde{w}_k, \widetilde{a}_k$) such that,

- when multiplying $\mathbb{1}_{\langle \widetilde{w}_k, h_{i-1} \rangle + \langle \widetilde{a}_k, x_i \rangle \geq 0}$ gives the target $\Phi_{i \to j, r, s}(\langle w_{i \to j, r, s}^*, x_i^\star \rangle)$, but
- when multiplying $\mathbb{1}_{\langle \widetilde{w}_k, h_{i'-1} \rangle + \langle \widetilde{a}_k, x_{i'} \rangle \geq 0}$ gives near zero.

The full proof is in Appendix G.2 but we sketch why Lemma 5.2 can be derived from Lemma 5.1.

*Sketch proof of Lemma 5.2.* Let us focus on indicator $\mathbb{1}_{\langle \widetilde{w}_k, h_{i'-1} \rangle + \langle \widetilde{a}_k, x_{i'} \rangle \geq 0}$:

- $\langle \widetilde{a}_k, x_{i'} \rangle$ is distributed like $\mathcal{N}(0, \frac{2\varepsilon_x^2}{m})$ because $\langle \widetilde{a}_k, x_{i'} \rangle = \langle (\widetilde{a}_k, (\varepsilon_x x_{i'}^\star, 0) \rangle$; but
- $\langle \widetilde{w}_k, h_{i'-1} \rangle$ is roughly $\mathcal{N}(0, \frac{2}{m})$ because $\|h_{i'-1}\| \approx 1$ by random init. (see Lemma B.1a).

Thus, if we treat $\langle \widetilde{w}_k, h_{i'-1} \rangle$ as the "noise $n$" in Lemma 5.1 it can be $\frac{1}{\varepsilon_x}$ times larger than $\langle \widetilde{a}_k, x_{i'} \rangle$.

To show Lemma 5.2a, we only need to focus on $|\langle \widetilde{w}_k, h_{i'-1}^{(0)} \rangle| \leq \frac{\varepsilon_c}{\sqrt{m}}$ because $i = i'$. Since $h^{(0)}$ can be shown close to $h$ (see Lemma D.1), this is almost equivalent to $|\langle \widetilde{w}_k, h_{i'-1} \rangle| \leq \frac{\varepsilon_c}{\sqrt{m}}$. Conditioning on this happens, the "noise $n$" must be small so we can apply Lemma 5.1a.

To show Lemma 5.2a, we can show when $i' \neq i$, the indicator on $|\langle \widetilde{w}_k, h_{i-1} \rangle| \leq \frac{\varepsilon_c}{\sqrt{m}}$ gives *little* information about the true noise $\langle \widetilde{w}_k, h_{i'-1} \rangle$. This is so because $h_{i-1}$ and $h_{i'-1}$ are somewhat uncorrelated (details in Lemma B.1k). As a result, the "noise $n$" is still large and thus Lemma 5.1b applies with $\Phi_{i \to j, r, s}(0) = 0$. $\qquad\square$

## 5.3 Fitting the Target Function

We are now ready to design $W^\circledast \in \mathbb{R}^{m \times m}$ using Lemma 5.2.

**Definition 5.3.** *Suppose* $\varepsilon_e \in \left( 0, \frac{1}{\mathfrak{C}_\mathfrak{s}(\Phi, O(\sqrt{L}))} \right)$, $C' = \mathfrak{C}_{\varepsilon_e}(\Phi, \sqrt{L})$, $\varepsilon_x \in (0, \frac{1}{\rho^4 C'})$, *we choose*

$$\varepsilon_c := \frac{\varepsilon_e \varepsilon_x}{4C'} \ , \quad C := \frac{4(C')^2}{\varepsilon_e \varepsilon_x} \ , \quad C_{i \to j, s} := \frac{1}{m} \left\| \mathbf{e}_s^\top \mathsf{Back}_{i \to j}^{(0)} \right\|_2^2 \|h_{i-1}^{(0)}\|^2 \ .$$

*We construct* $W^\circledast \in \mathbb{R}^{m \times m}$ *by defining its $k$-th row vector as follows:*

$$w_k^\circledast := \sum_{i=2}^{L-1} \sum_{j=i+1}^{L} \sum_{r \in [p], s \in [d]} \frac{1}{mC_{i \to j, s}} \left[ \mathbf{e}_s^\top \mathsf{Back}_{i \to j}^{(0)} \right]_k \mathbb{1}_{|\langle w_k, h_{i-1}^{(0)} \rangle| \leq \frac{\varepsilon_c}{\sqrt{m}}} H_{i \to j, r, s}(a_k) h_{i-1}^{(0)}$$

$$\text{where} \quad C_{i \to j, s} := \frac{1}{m} \left\| \mathbf{e}_s^\top \mathsf{Back}_{i \to j}^{(0)} \right\|_2^2 \|h_{i-1}^{(0)}\|^2$$

*Above, functions* $H_{i \to j, r, s} \colon \mathbb{R} \to [-C, C]$ *come from Lemma 5.2.*

The following lemma that says $f_{j', s'}$ is close to the target function $F_{j', s'}^*$.

**Lemma 5.4** (existence through backward). *The construction of $W^\circledast$ in Definition 5.3 satisfies the following. For every normalized input sequence $x$ generated from $x^\star$ in the support of $\mathcal{D}$, we have with probability at least $1 - e^{-\Omega(\rho^2)}$ over $W, A, B$, it holds for every $3 \leq j' \leq L$ and $s' \in [d]$*

$$f_{j', s'} = \sum_{i=2}^{j'-1} \sum_{r \in [p]} \Phi_{i \to j', r, s'}(\langle w_{i \to j', r, s'}^*, x_i^\star \rangle) \pm \left( p\rho^{11} \cdot O(\varepsilon_e + \mathfrak{C}_\mathfrak{s}(\Phi, 1)\varepsilon_x^{1/3} + Cm^{-0.05}) \right) \ .$$

*Proof sketch of Lemma 5.4.* Using definition of $f_{j',s'}$ in (5.1) and $W^*$, one can write down

$$f_{j',s'} = \sum_{i',j',j} \sum_{r\in[p],s\in[d]} \sum_{k\in[m]} \left( \frac{1}{mC_{i\to j',s}} \left[ \mathbf{e}_{s'}^\top \, \mathsf{Back}_{i'\to j'} \right]_k \left[ \mathbf{e}_s^\top \, \mathsf{Back}_{i\to j}^{(0)} \right]_k \right.$$

$$\left. \times \mathbb{1}_{|\langle w_k, h_{i-1}^{(0)}\rangle| \le \frac{\varepsilon_c}{\sqrt{m}}} \mathbb{1}_{[g_{i'}]_k \ge 0} H_{i\to j,r,s}(a_k)\langle h_{i-1}, h_{i-1}^{(0)}\rangle \right) \qquad (5.3)$$

Now,

- The summands in (5.3) with $i \ne i'$ are negligible owing to Lemma 5.2b.
- The summands in (5.3) with $i = i'$ but $j \ne j'$ are negligible, after proving that $\mathsf{Back}_{i\to j}$ and $\mathsf{Back}_{i\to j'}$ are very uncorrelated (details in Lemma C.1).
- The summands in (5.3) with $s \ne s'$ are negligible using the randomness of $B$.
- One can also prove $\mathsf{Back}_{i'\to j'} \approx \mathsf{Back}_{i'\to j'}^{(0)}$ and $h_{i'-1} \approx h_{i'-1}^{(0)}$ (details in Lemma D.1).

Together,

$$f_{j',s'} \approx \sum_{i'=2}^{j'-1} \sum_{r\in[p]} \sum_{k\in[m]} \left( \frac{1}{mC_{i'\to j',s'}} \left( \left[ \mathbf{e}_{s'}^\top \, \mathsf{Back}_{i'\to j'}^{(0)} \right]_k \right)^2 \right.$$

$$\left. \cdot \mathbb{1}_{|\langle w_k, h_{i'-1}^{(0)}\rangle| \le \frac{\varepsilon_c}{\sqrt{m}}} \mathbb{1}_{[g_{i'}]_k \ge 0} H_{i'\to j,r,s'}(a_k) \| h_{i'-1}^{(0)} \|^2 \right)$$

Applying Lemma 5.2a and using our choice of $C_{i'\to j',s'}$, this gives (in expectation)

$$f_{j',s'} \approx \sum_{i=2}^{j'-1} \sum_{r\in[p]} \Phi_{i\to j',r,s'}(\langle w_{i\to j',r,s'}^*, x_i^\star\rangle) = F_{j',s'}^*(x^\star) \ .$$

Proving concentration (with respect to $k \in [m]$) is a lot more challenging due to the sophisticated correlations across different indices $k$. To achieve this, we replace some of the pairs $w_k, a_k$ with fresh new samples $\widetilde{w}_k, \widetilde{a}_k$ for all $k \in \mathcal{N}$ and apply concentration only with respect to $k \in \mathcal{N}$. Here, $\mathcal{N}$ is a random subset of $[m]$ with cardinality $m^{0.1}$. We show that the network stabilizes (details in Section E) against such re-randomization. Full proof is in Section G.3. □

Finally, one can show $\|W^*\|_F \le O\left(\frac{p\rho^3 C}{\sqrt{m}}\right)$ (see Claim G.1). Crucially, this Frobenius norm scales in $m^{-1/2}$ so standard SGD analysis shall ensure that our sample complexity does not depend on $m$ (up to log factors).

## Footnotes

*Full version and future updates can be found on `https://arxiv.org/abs/1902.01028`.

[2]For instance, if $W \in \mathbb{R}^{m \times m}$ is the recurrent weight matrix, and is followed with an ReLU activation $\sigma$. Under standard random initialization $\mathcal{N}(0, \frac{2}{m})$, the combined operator $\sigma(Wx) \colon \mathbb{R}^m \to \mathbb{R}^m$ has operator norm $\sqrt{2}$ with high probability. If instead one uses $\mathcal{N}(0, \frac{\sqrt{2}}{m})$, then $\beta$ becomes 1 but gradients will vanish exponentially fast in $L$.

[3]This is without loss of generality, since $\frac{1}{2}$ can always be padded to the last coordinate, and $\|x_\ell^\star\|_2 = 1$ can always be ensured from $\|x_\ell^\star\|_2 \leq 1$ by padding $\sqrt{1 - \|x_\ell^\star\|_2^2}$ to the second-last coordinate. We make this assumption to simplify our notations: for instance, $(x_\ell^\star)_{d_x} = \frac{1}{2}$ allows us to focus only on networks in the concept class without bias.

[4]We use $[-1, 1]$ and 1-Lipschitzness for notation simplicity. In generally, our final time and sample complexity bounds only scale polynomially with the boundedness and Lipschitzness parameters.

[5]Equivalently, one can scale matrix $B$ by factor $\lambda$. For notational simplicity, we split the matrix into $W + W'$ but this does not change the algorithm since gradient with respect to $W + W'$ is the same with respect to $W'$.

[6]More specifically, Allen-Zhu, Li, and Liang [1] study two (or three) layer feedforward networks, which use one (or two) hidden weight matrix to learn one target function. Here in RNN, there is only *one* weight matrix shared across the entire time horizon to learn $L$ target functions at different input positions. In other words, using "one weight" to learn "one target function" is known from prior work, but using "one weight" to efficiently learn "$L$ different target functions" is substantially more difficult, especially when the position information is not given as the input to the network. For example, our theorem implies that an RNN can *distinguish* the sequences "AAAB" from "AABA", since the order of A and B are different. This requires the RNN to keep track, *using one weight matrix*, of the position information of the symbols in the sequence.

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
