[Reviews · NeurIPS 2019]

Reviewer 1



The paper is interesting and notable for tackling the difficult case of using recurrent neural networks to do learning. The function class it learns seems to be identical to the function classes learned by Allen-Zhu et al. in prior works, which are known to be learnable in polynomial-time via other methods. The main technical difference from this paper and previous work (per the authors) seems to be Lemma 5.2b. What happens if the inputs are not norm 1? What sort of complexity bounds are obtained in this case? Also, how does the complexity scale with the lipschitzness/boundedness of the loss function (currently assumed to be 1)? Are there hardness results indicating that your results are best possible? For example, the 'almost polynomial' result of L^{O(log 1/\eps)}, is it possible this can be improved? Can you give some explanation as to what this function H is from Lemma 5.2? Roughly what techniques are involved? Why is "n" considered a noise parameter? I did not follow. The main concern is how much does this paper overlap with several other works analyzing SGD on overparameterized networks, most notably the work of Allen-Zhu et al. The authors indicate that the proof here is substantially different and more difficult.

Reviewer 2



This paper show that Elman RNNs optimized with vanilla SGD can learn concepts where the target output at each position of the sequence is any function of the previous L inputs that can be encoded in a two-layer smooth neural network. There are multiple assumptions and complications in showing the main result. The crux of the proof is to show that if the RNN is overparameterized enough, then if we start from a randomly initialized RNN matrix W, there exists a function which is linear in matrix W* whose value at a specific W* is a good approximation to the target in the concept class. Showing that SGD moves in a direction similar to such W* gives the desired result. Another interesting aspect of the main result is that the number of samples that SGD needs depends only logarithmically with respect to the number of RNN neurons, making it applicable to overparameterized settings. Indeed, for the result to hold, the number of RNN neurons must depend polynomially (or slightly more depending on the complexity of the concept class) on the length L of the sequences. Pros: - Novel result for RNN learnability with generalization bound polynomial in input length. - Sample complexity bound almost independent of size of the RNN - Different properties of RNNs at random initialization (not sure which of these are novel when compared to [2]) Cons: - RNN is so overparameterized that there's a quick shortcut leading close to the optimal solution - Similar in vein to [2] Other: - Can you give some concrete (possibly practical) examples of functions belonging to the concept class? For example does the concept class allow to count, or, say, determine whether an a^nb^m sequence is such that n=m? - Can you give an intuition on the function complexities C_\eps and C_s?

Reviewer 3



The paper proves that RNN trained via SGD when only the weights are trained converges to a point close to the optimum and that the number of examples required for training scales polynomially with the number of training examples. The result is very interesting and novel. Yet, I have few remarks: 1. The work "Robust Large Margin Deep Neural Networks" provides generalization error guarantees that are independent of the number of neurons, unlike what is written in the paper that there is no such generalization result for neural networks till now. 2. I would appreciate seeing a simulation that demonstrates the result proposed in the paper. I believe it is not hard to train such a network when only one matrix is optimized. Indeed, as only one parameter is optimized the performance will be poor but still, it would demonstrate the result. 3. I would appreciate a discussion as to how the work may be extended to learning of the other parameters.

[Author Response · NeurIPS 2019]

We thank all the reviewers for the time reading our paper! We will fix all the minor issues, and below we only address
the main concerns.

• **R2:** "The function class it learns seems to be identical to the function classes learned by Allen-Zhu et al. in prior
works, which are known to be learnable in polynomial-time via other methods."

We'd like to emphasize that our main focus of the paper is "what function class can *recurrent network (RNN)* learn".
Prior to our work, the only function class known to be learnable are linear functions; but even for this linear class, it
could require $2^L$ samples if known generalization bounds are used, since $\|W\|_2$ is larger than 1 (roughly 2 in our
setting). We believe it is (much) more interesting if someone could identify a non-linear class and prove that it is
learnable by RNN. This is already a step forward.

• **R2:** What happens if the inputs are not norm 1? What complexity bounds are obtained in this case? Also, how does
the complexity scale with the Lipschitzness/boundedness of the loss function (currently assumed to be 1)?

If the inputs are not norm 1 but with norm $< C$, then we can wlog. pad the input (adding $\sqrt{1-C^2}$ to last
coordinate) to make it norm exactly C. Then, the concept class $\phi(\langle w, x\rangle)$ becomes $\phi(C\langle w, x/C\rangle)$ and one can
define $\phi'(z) := \phi(Cz)$. Now, the complexity changes from $\mathfrak{C}(\phi, R)$ to $\mathfrak{C}(\phi', R)$: how much it affects the complexity
depends on what $\phi$ is. If $\phi$ is degree $k$ then this is at most $C^k$. If $\phi(z) = \sin(z)$ and $\phi'(z) = \sin(Cz)$, this changes
the complexity by a factor $(1/\varepsilon)^C$. This is polynomial only if $C$ is a constant, but should be somewhat forgiven:
learning $\sin(10\langle w, x\rangle)$ is indeed somewhat "exponentially harder" than learning $\sin(\langle w, x\rangle)$.

As for the Lipschitzness/boundedness of the loss function, our final result (time/sample complexities) only scale
polynomially with them. We will add a short section explaining this.

• **R2:** Are there hardness results? is $L^{O(\log 1/\varepsilon)}$ the best possible?

There's no hardness result, but we conjecture $L^{O(\log 1/\varepsilon)}$ is the best possible for vanilla RNN. If more structures
such as memory units are added, it may be possible to get $poly(L)$. That's an interesting future direction.

• **R2:** My main concern is how much does this paper overlap with several other works analyzing SGD on overparame-
terized networks, most notably Allen-Zhu et al.

Allen-Zhu et al. [1] consider two (or three) layer feedforward NN, which uses one (or two) hidden weight matrix to
learn one target function. Here in RNN, there's only **one** weight matrix shared across the entire time horizon to
learn $L$ target functions at different input positions.

Using "one weight" to learn "one target function" is traditional,
but using "one weight" to efficiently learn "$L$ different target functions" is substantially more difficult,

especially when the layer information $L$ is not given as the input to the network. For example, our theorem implies
that an RNN can **distinguish** the sequences "AAAB" from "AABA", since the order of A and B are different.
This requires the RNN to keep track, **using one weight matrix**, of the position information of the symbols in the
sequence. This is indeed is more difficult.

Allen-Zhu et al. [2,3] are about trainability only, and gives no generalization guarantee.

• **R2:** the main technical difference from previous work seems to be Lemma 5.2b.

**No, this is not true.** Due to space, we only present one of our main contributions in 8 pages. As we emphasized
on line 269-273 of page 8. Our technical lemmas are in Appendices B/C/D/E/F (regarding RNN at initialization
and stability). Although B+F have reused some prior work, Appendices C+D+E are **completely new** what-so-ever.
Furthermore, C+D+E give technical lemmas that are (certainly) of independent interests.

• **R4** also questions our overlap with Allen-Zhu et al [2]. We refer R4 to our answers above.

• **R4:** Can you give examples about the proposed concept class? Intuitions about $\mathfrak{C}_s$ and $\mathfrak{C}_e$?

**Counting is an example.** One can define $\phi$ such that $\phi(a) = 1$ and $\phi(b) = -1$ (this can be achieved by a quadratic
function with constant complexity), hence the final prediction (target function) is $\sum_i \phi(x_i)$. If the sequence is
$x = a^n b^m$ such that $n = m$, then $\sum_i \phi(x_i) = 0$, otherwise it is non-zero. So this concept class is learnable by our
theorem. We will add more examples in the next version. We will also give more intuitions about $\mathfrak{C}_s$ and $\mathfrak{C}_e$.

• **R6:** "Robust Large Margin Deep Neural Networks" gives generalization independent of number of neurons

First, "Robust..." seems to be about feedforward NN and not RNN. Second, as we have articulated in lines 35-55,
there are indeed RNN generalization works [31,9] that do not depend on number of neurons, but they have to depend
exponentially on input length.

• **R6:** what will happen if other parameters are learned? Our same result will hold (almost no proof changes) if
$A, B, W$ are all trained together. We will add a paragraph to explain it.

[Meta-Review · NeurIPS 2019]

This paper provides theory that explains what functions can be learned using RNNs (beyond linear classifiers) and gives sample complexity bounds. All reviewers agree that this result is significant and therefore, I recommend acceptance.